# Natural Oscillatory Frequency Slowing in the Premotor Cortex of Early-Course Schizophrenia Patients: A TMS-EEG Study

**DOI:** 10.3390/brainsci13040534

**Published:** 2023-03-24

**Authors:** Francesco L. Donati, Ahmad Mayeli, Kamakashi Sharma, Sabine A. Janssen, Alice D. Lagoy, Adenauer G. Casali, Fabio Ferrarelli

**Affiliations:** 1Department of Psychiatry, University of Pittsburgh, 3501 Forbes Avenue, Suite 456, Pittsburgh, PA 15213, USA; 2Western Psychiatric Hospital, University of Pittsburgh Medical Center, Pittsburgh, PA 15213, USA; 3Department of Health Sciences, University of Milan, 20142 Milan, Italy; 4Institute of Science and Technology, Federal University of São Paulo, São José dos Campos 12231-280, Brazil

**Keywords:** schizophrenia, early-course, TMS, TMS-EEG, premotor, oscillations

## Abstract

Despite the heavy burden of schizophrenia, research on biomarkers associated with its early course is still ongoing. Single-pulse Transcranial Magnetic Stimulation coupled with electroencephalography (TMS-EEG) has revealed that the main oscillatory frequency (or “natural frequency”) is reduced in several frontal brain areas, including the premotor cortex, of chronic patients with schizophrenia. However, no study has explored the natural frequency at the beginning of illness. Here, we used TMS-EEG to probe the intrinsic oscillatory properties of the left premotor cortex in early-course schizophrenia patients (<2 years from onset) and age/gender-matched healthy comparison subjects (HCs). State-of-the-art real-time monitoring of EEG responses to TMS and noise-masking procedures were employed to ensure data quality. We found that the natural frequency of the premotor cortex was significantly reduced in early-course schizophrenia compared to HCs. No correlation was found between the natural frequency and age, clinical symptom severity, or dose of antipsychotic medications at the time of TMS-EEG. This finding extends to early-course schizophrenia previous evidence in chronic patients and supports the hypothesis of a deficit in frontal cortical synchronization as a core mechanism underlying this disorder. Future work should further explore the putative role of frontal natural frequencies as early pathophysiological biomarkers for schizophrenia.

## 1. Introduction

Schizophrenia is a major psychiatric condition that ranks among the leading causes of disability worldwide [1]. Illness onset typically occurs between 15 to 30 years of age with the emergence of positive (hallucinations, delusions, disorganized thought/behavior) and negative (anhedonia, avolition, flattened affect) symptoms that are accompanied by a progressive decline in cognitive and social abilities. This functional decline is particularly marked in the first years after the first psychotic break; accordingly, the early course of schizophrenia has been regarded as an important window for intervention [2] and tertiary prevention (the so-called “critical period” hypothesis, [3]). In this light, the search for objective biomarkers that can be reliably detected since the earliest stages of schizophrenia has been a major focus in psychiatric research due to their potential to inform novel treatment strategies [4]. Early biomarkers of schizophrenia are also more likely to be directly linked to its pathophysiology rather than to superimposed confounding factors such as duration of psychosis, length of exposure to antipsychotic medications, neurodegenerative processes, and comorbidities, and may thus enable a better understanding of the biological underpinning of chronic psychoses [5].

Transcranial magnetic stimulation (TMS) has increasingly gained popularity in psychiatric research and therapeutics, given its ability to precisely and non-invasively target cortical areas. When used repetitively, (r)TMS can effectively ameliorate conditions such as treatment-resistant depression and compulsive behaviors [6,7].

When coupled with electroencephalography (EEG), single-pulse TMS (TMS-EEG) may also serve as a probe to directly gauge brain function, examining the local excitability and the oscillatory properties of the targeted cortical area [8,9]. Furthermore, experimental paradigms coupling EEG with paired TMS pulses delivered at different intervals, such as short and long-interval-cortical-inhibition (SICI/LICI) [10,11,12], or combining TMS-EEG and peripheral somatosensory stimulation (such as short-latency afferent inhibition, SAI) [13,14], have been employed to elucidate cortical mechanisms of inhibitory control in the healthy brain (for a comprehensive review, see [15]).

Among others, a prominent TMS-EEG measure of local cortical excitability is the main oscillatory frequency that can be recorded for a given cortical area, or natural frequency. Much like the chords of a musical instrument are each tuned to a specific frequency, different brain areas respond with different natural frequencies when stimulated with TMS [16]. Specifically, in healthy individuals, the prefrontal natural frequency is in the gamma frequency band, the premotor natural frequency is in the high beta/low gamma range, the parietal natural frequency is in the beta band, and the occipital natural frequency is in the alpha range [16,17]. These findings were recently replicated in a study investigating TMS-EEG natural frequencies from both the right and left hemisphere [18] as well as with concurrent EEG and intracranial stimulation and recordings [19].

Several studies have employed TMS-EEG to study brain function in schizophrenia. Work from our and other research groups has shown that, in chronic patients with schizophrenia, the TMS-evoked EEG potentials (TEPs) were reduced when TMS was delivered to the premotor cortex [20] or the vertex (i.e., Cz electrode) [21]. TMS-EEG studies employing the LICI paradigm have demonstrated that cortical inhibition is reduced in the dorsolateral prefrontal cortex (DLPFC) of patients with schizophrenia [22,23], particularly for activity in the gamma frequency band [24]. One recent study applying the SAI paradigm to both the primary motor area and the DLPFC of patients with schizophrenia found that the peripheral nerve stimulation preceding the TMS pulse elicited alterations at different latencies of the TEP, and that these alterations were different between the two brain areas [25]. Interestingly, only deficits in DLPFC-SAI correlated with behavior (i.e., deficits in executive function).

Our research group has previously shown that patients with chronic schizophrenia exhibited a loss of the rostro-caudal gradient of natural frequencies seen in the healthy brain [17]. Specifically, areas of the frontal lobe, including the DLPFC, motor, and premotor cortices, showed a marked slowing in the natural frequencies of these patients. The maximal reduction was found in the DLPFC, followed by the premotor cortex, suggesting a progressive local disruption of synchronized neuronal firing in the frontal lobe. However, whether altered natural frequencies in these anterior frontal areas are present in early stages of schizophrenia remains to be determined.

While a dysfunction of the DLPFC is central to the pathophysiology of schizophrenia [26], the antero-lateral location of this brain area poses technical challenges to its investigation with TMS-EEG, primarily related to the interposition of cranial muscles between the TMS coil and the cortical target [27,28]. The resulting muscle activation often leads to large EEG artifacts superimposed to the TEP, which may require additional signal processing [29]. In turn, this may affect the early components of the TEP, which represent an important readout of an effective cortical stimulation [30]. This possibly explains the low amplitude of the early TEP components in some previous studies targeting the prefrontal cortex, including some studies in schizophrenia [23,25]. These limitations may reduce the potential for translation to the clinical setting of prefrontal TMS-EEG biomarkers for schizophrenia. Conversely, a more centro-medial area such as the premotor cortex is generally less affected by muscle activation and related artifacts [31,32] (see also Figure 6 in [27]).

In this study, we used TMS-EEG to probe the intrinsic oscillatory properties, including the natural frequency, of the left premotor cortex in *N* = 16 patients with early-course schizophrenia and *N* = 16 age and gender-matched healthy control subjects. We hypothesized that the premotor natural frequency would be reduced in the patients’ group, reflecting a local deficit in synchronization that is present since the early stages of schizophrenia.

## 2. Materials and Methods

### 2.1. Participants

We recruited sixteen patients with early-course schizophrenia (ECSCZ, defined as <2 years from a first psychotic episode, as in previous studies [33]) and sixteen healthy control subjects (HCs). Common exclusion criteria included major medical or neurological conditions affecting the central nervous system, intellectual disability according to DSM-5 criteria, pregnancy or postpartum, and inability to complete magnetic resonance imaging (MRI) scans or TMS. Exclusion criteria for HCs included a history of treatment with antipsychotic medications; personal or family history of schizophrenia-spectrum disorder or psychosis; and current use of psychotropic medications.

Table 1 summarizes the demographics of the study populations. No significant differences in age and gender were found between groups. Subjects were evaluated by one expert rater, and the diagnosis of schizophrenia was confirmed with the Structured Clinical Interview for DSM Disorders (SCID) [34]. The severity of psychotic symptoms in the patients’ group was quantified using the Scale for the Assessment of Positive Symptoms (SAPS) and the Scale for the Assessment of Negative Symptoms (SANS) (Table 1).

Each participant underwent structural MRI acquisition to be used for TMS neuronavigation.

The study protocol was approved by the University of Pittsburgh Institutional Review Board, and all participants provided written informed consent prior to completing study procedures.

### 2.2. Procedure

Study participants sat comfortably on a reclining chair. For the assessment of each subject’s resting motor threshold (RMT), the left motor cortex was identified on T1-weighted individual MRIs using a neuronavigation system (Localite Classic Edition, Bonn, Germany). Following international guidelines, the RMT was determined as the lowest intensity capable of eliciting an electromyographic response of the *abductor pollicis brevis* muscle > 50 μV in 5 out of 10 TMS trials. [35]. TMS was delivered in biphasic single-pulses using a TMS stimulator (MagPro X100, MagVenture, Farum, Denmark) and a figure of 8 coil (MagVenture MCF-B65). The RMT, expressed as % of Maximum Stimulator Output (%MSO), did not significantly differ between the two groups (HC: 51.0 ± 5.38; ECSCZ: 55.75 ± 10.44; *p* = 0.1305 Wilcoxon rank-sum test).

Then, the neuronavigation target was moved to the left middle frontal gyrus and adjusted to match the coordinates [x: −26; y: −4; z: 69] of the Montreal Neurological Institute (MNI) space [36]. This targeting spot lies in the left Brodmann Area 6 (BA6), i.e., the left premotor cortex. The intensity of stimulation for TMS-EEG at this target was set to 120% of the RMT, with a stimulation angle of 45 degrees relative to the midline.

We used a 64-electrode cap based on the 10–20 system (Easycap), passive ring-shaped EEG electrodes, and a TMS-compatible amplifier (BrainAmp DC, Brain Products, Gilching, Germany). The quality of the TMS-evoked EEG response was ensured by employing an online, real-time graphical interface displaying the signal from averaged trials referenced to the average of all channels [37]. Briefly, before the EEG recording was started, the TEP resulting from 20 TMS pulses was visualized. The EEG recording was started only if (1) no large-amplitude (>100 μV) decay or muscle-related artifacts were visualized and (2) the first component of the average TEP had an amplitude >5 μv. Otherwise, the coil orientation and/or the TMS target were slightly adjusted, and the real-time inspection was repeated. A freely available brain atlas based on MNI coordinates was used to confirm that the final targeting spot lied on BA6.

EEG data were acquired at a sampling rate of 5000 Hz. Each recorded TMS-EEG session consisted of 150 single pulses with a jittered interstimulus interval (0.4 to 0.6 Hz), a frequency similar to those employed by previous studies probing the premotor cortex [16,38,39,40,41,42] and which does not induce significant neuronal plasticity in BA6 [43]. To reduce the amplitude of “off-target” (i.e., stereotypically induced regardless of the stimulation site) auditory artifacts due to the TMS “click” sound and contaminating the TMS-evoked EEG responses [44], participants wore earbuds playing a state-of-the-art, TMS-specific noise masking track [38].

### 2.3. Data Analysis

Data analysis was performed with Matlab R2017a (The Mathworks, Natick, MA, USA) employing customized algorithms based on the EEGlab toolbox [45] and the SiSyPhus Project interface (SSP 2.8U, University of Milan, Italy), as in previous studies [46,47]. After the removal of channels contaminated by noise (“bad” channels, HC: 4.31 ± 3.11; ECSCZ: 6.43 ± 6.34; *p* = 0.7332; Wilcoxon rank-sum test), EEG signals were split into trials of 1600 ms around the TMS pulse (−800 +800 ms, time 0 corresponding to the TMS pulse). Trials contaminated by noise, artifacts from eye movements, or muscle activity were rejected by visual inspection, which resulted in a comparable number of remaining EEG trials between the two groups (HC: 135.68 ± 11.53; ECSCZ: 136.68 ± 14.91; *p* = 0.6371; Wilcoxon rank-sum test).

Artifacts from the high-energy TMS pulse were removed from each trial by replacing the interval around the pulse (from −2 ms to 6 ms) with the data immediately before (from −10 ms to −2 ms). A fifth-order moving-average filter was applied between 4 and 8 ms to reduce high-frequency edges. EEG signals were downsampled to 1000 Hz, bandpass filtered (1–80 Hz), re-referenced to the average of all channels, and baseline corrected.

Residual artifacts from eye movements, muscle activity, cardiac, and TMS pulse were excluded using Independent Component Analysis (ICA). Only components clearly imputable to artifacts were removed. The spherical function of the EEGLAB toolbox [45] was then used to interpolate bad channels. In order to quantify the TMS-evoked EEG response in the time domain, the Mean Field Power (MFP, [48]) was calculated across all channels (Global MFP, GMFP) as well as by averaging the voltages squared across the channels surrounding the stimulation site (FC5, FC3, FC1, FCz, C5, C3, C1, Cz; Local MFP, LMFP).

A Morlet time-frequency decomposition (3.5 cycles) was employed to analyze TMS-evoked oscillatory activity. Event-related spectral perturbation (ERSP) matrices between 8 and 45 Hz were computed as the ratio of the spectral power (μV^2^) of individual EEG trials and their respective mean baseline spectra. These parameters (wavelet cycles, frequency interval) were chosen to maximize the comparability of findings with previous studies investigating the natural frequency of the premotor cortex [16,17,42].

After computing the average across trials, time-frequency matrices of significant ERSP values were extracted. This was achieved by using a two-tailed bootstrap significance probability level with respect to the baseline, which was constructed from 500 permutations and applied to the full-epoch window (α < 0.05 after False Discovery Rate correction for multiple comparisons). The ERSP was then averaged over frequency in alpha (8–12 Hz), beta (13–30 Hz), and gamma (30–45 Hz) bands and then compared between groups across all channels and in the cluster of channels surrounding the stimulation site. Finally, we extracted the EEG frequency with the highest cumulate power calculated at the electrode closest to the stimulation site, i.e., the natural frequency, as in previous work [17].

### 2.4. Statistical Analysis

A two-tailed t-test was used to compare age between groups; χ-squared tests were used to assess differences between groups for dichotomous variables (i.e., gender). Wilcoxon-rank sum tests were used to establish statistical differences in TMS-EEG measures between early-course schizophrenia patients and HC participants. In patients, Spearman’s correlation coefficients were calculated between TMS-EEG measures and SAPS/SANS scores and dose of antipsychotic medications, quantified as chlorpromazine equivalents (see Table 1). The level of significance was set at 0.05 for all tests.

## 3. Results

In both groups, single-pulse TMS determined several early (<100 ms) voltage deflections >5 μv (Figure 1), which is consistent with effective cortical stimulation [30,49]. In addition, the relatively low amplitude of late (>100 ms) TEP components suggested an effective suppression of “off-target” auditory artifacts [38,44].

No significant differences were found when comparing the time course of the EEG responses to TMS, as quantified by GMFP and LMFP, across groups (GMFP, HC 0.77 ± 0.69 μV; ECSCZ: 0.61 ± 0.43 μV; *p* = 0.6375. LMFP, HC: 0.92 ± 0.91 μV; ECSCZ: 0.56 ± 0.42 μV; *p* = 0.2662. Wilcoxon rank-sum tests). Appendix A show grand averages of GMFP and LMFP and time-bin-by-time-bin group comparisons (no significant differences after Wilcoxon rank-sum tests).

In the time-frequency domain, no differences in ERSP were found between groups in any frequency band, both in a region of interest (ROI) overlying the left premotor cortex and across all EEG channels (see Appendix A). However, the natural frequency evoked by the stimulation of the left premotor cortex was significantly reduced in ECSCZ patients (HC: 29.15 ± 6.59 Hz; ECSCZ: 23.27 ± 4.27 Hz; *p* = 0.0186; Wilcoxon rank-sum test), corresponding to a large effect size (Cohen’s d = 1.06). Individual values and boxplots with median, quartiles, and extreme values for both groups are shown in Figure 2.

No correlation was found in either group between the natural frequency and age. In ECSCZ patients, no significant correlations were found between the natural frequency and the patients’ positive or negative symptoms quantified by SAPS (ρ = −0.06; *p* = 0.8245) and SANS (ρ = −0.28; *p* = 0.2882) scores, respectively, nor with the dose of antipsychotic medications that the patients were taking at the time of the TMS-EEG recordings (ρ = −0.27; *p* = 0.3679).

## 4. Discussion

In this study, we performed single-pulse TMS-EEG of the left premotor cortex in patients with early-course schizophrenia and in HC subjects (Figure 1). Our main finding was a significant reduction in the natural frequency of patients with schizophrenia compared to HCs (Figure 2), corroborating the hypothesis that a slowing of the natural oscillatory frequency of the premotor cortex is present not only in chronic schizophrenia, as previously reported by our group [17], but it is also a feature associated with the disorder since its earliest stages.

This finding is consistent with recent TMS-EEG evidence in first-episode psychosis, where the TMS-evoked EEG activity in the beta range was found to be reduced in another frontal area, the motor cortex [50]. Altogether, these results suggest that an impairment of the intrinsic oscillatory properties of frontal cortical circuits is likely a neurophysiological characteristic associated with the pathophysiology of schizophrenia rather than a superimposed alteration due to neurodegenerative processes, duration of illness, and/or length of exposure to antipsychotic medications.

Deficits in fast frontal neural oscillations in patients with schizophrenia, including activity in the beta and gamma frequency bands, have been consistently replicated across different neurophysiological paradigms [51,52]. In generating these fast oscillations, an essential role is played by the inhibitory control exerted by parvalbumin+ gamma-aminobutyric acid (GABAergic) interneurons [53,54], a population of cells notably affected in schizophrenia [55,56]. In addition, GABAergic transmission is critically implicated in EEG responses to TMS [57]. Our findings are, therefore, consistent with a deficit in GABAergic control which is present since the early phases of the disorder. This expands previous findings from TMS studies using paired pulses in the motor cortex that demonstrated that deficits in inhibitory control are present in early-course schizophrenia [58,59], further supporting the hypothesis of an imbalance between cortical excitation and inhibition as a core mechanism underlying this disorder.

While several studies employing TMS-EEG in schizophrenia have successfully targeted the DLPFC [17,22,23,24,25], the technical challenges associated with investigating this cortical area (largely related to the interposition of lateral cranial muscles between the coil and the brain surface) may limit the potential for translation of the identified biomarkers [15,27,28,29,31,60]. This is particularly the case in the absence of a real-time TMS-EEG monitoring interface, which can be used to minimize muscle activation caused by TMS [27,30,37]. On the other hand, given the more centro-medial projection on the scalp of the premotor cortex, TMS-EEG recordings targeting this area are less affected by artifacts caused by muscle activation [31,32]. Here, we showed that the natural frequency of the premotor cortex is significantly reduced in early-course schizophrenia patients compared to HC subjects, with a large effect size (Cohen’s d = 1.06). However, it has to be noted that a certain overlap exists between the two distributions (Figure 2), which may limit the validity of this biomarker. Conversely, previous TMS-EEG studies targeting the DLPFC [17,25] have yielded to larger separation between chronic schizophrenia patients and HCs (up to complete non-overlap between groups [17]). Furthermore, the clinical significance of these DLPFC TMS-EEG biomarkers is supported by correlations with behavioral variables [17,25], which were not found in this study. Future research in early schizophrenia should investigate the natural frequency and other TMS-EEG measures at prefrontal sites and assess its clinical correlates. However, additional work should also elucidate the relationship between prefrontal and premotor TMS-EEG responses at the beginning of schizophrenia, aiming at the development of biomarkers for this disorder that are both accurate and feasible.

Future work is needed to address the limitations of the present study. First, we tested a relatively small cohort of patients. As increasing evidence has highlighted the large biological heterogeneity underlying the clinical phenotype of schizophrenia [61,62], our findings will need to be confirmed by future studies recruiting larger groups of patients. Second, we did not examine patients with other psychiatric conditions, including subjects with mood disorders. Importantly, one previous study has shown that the natural frequency of the premotor cortex is not only reduced in chronic schizophrenia but also during a depressive episode in subjects diagnosed with either major depressive disorder or bipolar disorder [42]. This suggests a shared deficit in cortical synchronization in the premotor cortex across these conditions, which warrants caution in the interpretation of our findings. More studies are therefore needed to elucidate whether an early deficit of premotor corticothalamic circuits is specific to schizophrenia or rather represents a window into cortical synchronization abnormalities that are shared between schizophrenia and mood disorders [63]. Third, our investigation was limited to the left hemisphere. Interestingly, one recent study in healthy subjects did not find significant differences in the natural frequencies between homologous areas of the left and right hemispheres, including between the left and right premotor cortices [18]. However, given the broad literature on altered lateralization of brain functions in schizophrenia [64,65,66], future studies should explore the symmetry of EEG responses to TMS in this disorder while investigating potential (lateralized) functional correlates (e.g., language, handedness). Finally, in the present study, we did not find any correlation between premotor TMS-EEG measures, including the natural frequency, and clinical symptoms. While this may suggest the absence of a relationship between a reduced intrinsic oscillatory activity of the premotor cortex and the clinical phenotype of schizophrenia (positive symptoms, negative symptoms), it is also possible that we were not powered enough to establish such relationships. Alternatively, the reduced intrinsic oscillatory activity in the premotor cortex may be related to behavioral alterations that were not investigated in our study, such as cognitive impairment or mirror neuron dysfunction. Thus, future studies administering cognitive tasks in larger groups of schizophrenia patients are needed to answer these questions.

## 5. Conclusions

The natural frequency of the premotor cortex is reduced since the early stages of schizophrenia, with a large effect size. As the interest for clinical applications of TMS-EEG arises, the premotor natural frequency may represent a feasible and inexpensive early pathophysiological biomarker for schizophrenia. However, more research is needed to elucidate its specificity, clinical significance, and relationship to the course of illness. As TMS is gaining momentum in psychiatric research and treatment, future studies should explore a role for TMS-EEG measures, including the natural frequency, in informing the diagnosis, prognosis, and personalization of treatment in schizophrenia and other psychiatric conditions [67,68].

## Figures and Tables

**Figure 1 brainsci-13-00534-f001:**
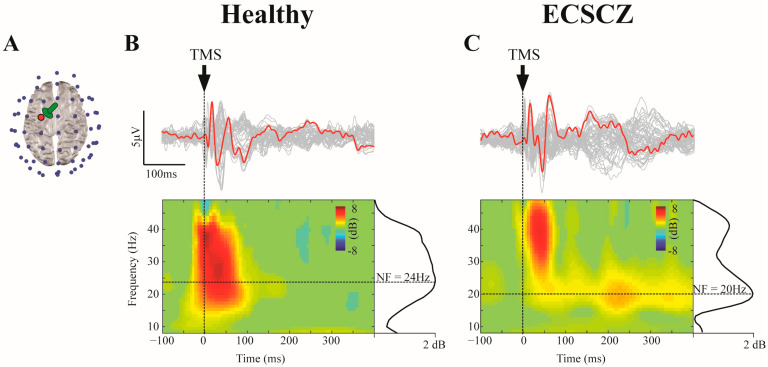
Single-pulse TMS of the left premotor cortex elicited EEG responses with similar time-courses, but slower natural frequencies in patients with early-course schizophrenia (ECSCZ) compared to HCs. Figure shows: (**A**) the position of the TMS coil (green) relative to the brain (gray) and the 64 EEG electrodes (blue dots). The red dot indicates the EEG electrode closer to the stimulation site, where the natural frequency was calculated. ((**B**), top) Butterfly plots of the TMS-evoked potential (TEP) averaged across trials in a representative HC subject. The red trace indicates the EEG electrode closest to the stimulation site. ((**B**), bottom) Event-Related Spectral Perturbation (ERSP) in the red electrode alongside the cumulate spectral power profile in the same electrode (solid black curves) and its peak frequency (or natural frequency, horizontal dotted lines). (**C**) depicts the same as in (**B**) in a representative ECSCZ patient. Vertical dotted lines under the arrows marked the time point when single-pulse TMS occurred (T0).

**Figure 2 brainsci-13-00534-f002:**
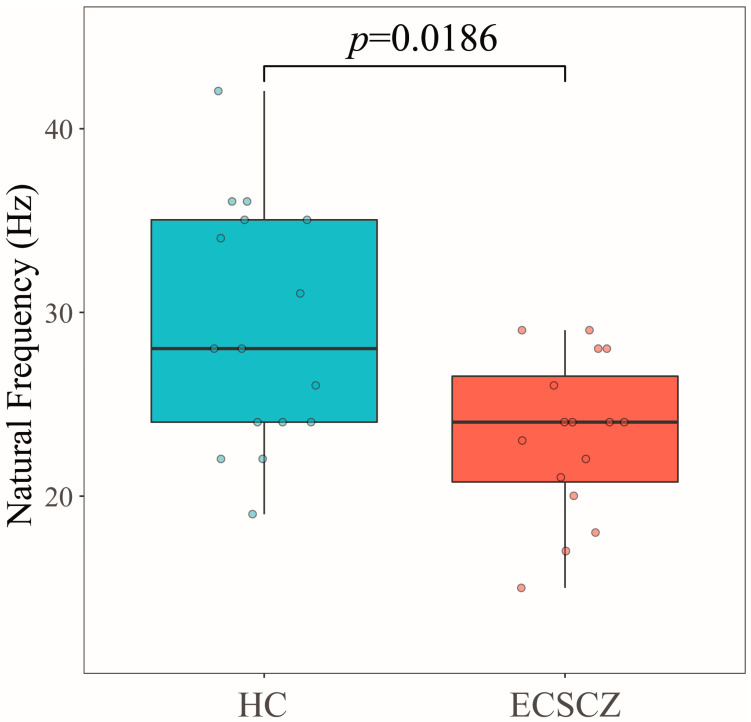
The natural frequency of the premotor area was significantly reduced in early-course schizophrenia compared to healthy control subjects. Figure shows boxplots with median, quartiles, and single-subject values for both groups, together with the *p*-value after Wilcoxon’s rank sum test.

**Table 1 brainsci-13-00534-t001:** **Study population demographics.** For the patients’ group, clinical rating scales and medication dose (expressed in chlorpromazine equivalents, CPZ eq) are reported. HC: healthy controls; ECSCZ: early-course schizophrenia patients; SAPS/SANS: Scale for the Assessment of Positive/Negative Symptoms.

	HC (*N* = 16)	ECSCZ (*N* = 16)	*p* Val
Age	23.43 ± 3.88	23.87 ± 4.29	0.7657
Gender	10 M; 6 F	11 M; 5 F	0.7097
SAPS	-	8.81 ± 6.99	-
SANS	-	32 ± 11.3	-
CPZ eq	-	331.16 ± 178.15	-

## Data Availability

The data presented in this study are available on request from the corresponding author.

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
