# Peer review of "Natural Oscillatory Frequency Slowing in the Premotor Cortex of Early-Course Schizophrenia Patients: A TMS-EEG Study"

_brainsci, 2023, doi:10.3390/brainsci13040534_

Round 1

Reviewer 1 Report

The current study combined single pulse TMS with EEG to investigate differences in TMS evoked oscillatory activity in the premotor cortex between a group of early (<2 years from onset) schizophrenic patients and age-matched healthy control participants. The authors report that the “natural frequency” in the left premotor cortex is reduced in early schizophrenia.

Overall, the writing of the paper feels rushed and is poorly developed. The introduction does not provide sufficient background. The discussion is overly brief and does not put the results into proper context. Finally, the methods are not clear, requiring the reader to infer several key methodological details. Major revisions are required.

Specific comments

Introduction

The authors fail to consider TMS-EEG work involving similar methodology. Work by Noda et al. (Schizophrenia Bulletin 2018) used a TMS measure called SAI to demonstrate reduced modulation of the N100 TEP elicited by TMS over DLPFC. This paper also demonstrated that the extent of modulation of the N100 was correlated with executive function.

The authors need to provide enhanced background over why they chose to target premotor cortex in this population. Their past work in chronic schizophrenia demonstrated reductions in the natural frequency in both premotor cortex and prefrontal cortex. However, the reductions were most pronounced in prefrontal cortex and there appeared to be a relationship with clinical scores/cognition with prefrontal natural frequency. As I note later, the authors briefly make a unsupported statement in the discussion that premotor cortex is more accessible for TMS-EEG investigations, but offer nothing to support this claim.

Methods    

Section 2.2 – No details are given about the TMS stimulus other than the site of stimulation, a procedure to ensure that largest TEP was elicited and the resting motor thresholds. What TMS model TMS stimulator, what model of coil, was the stimulation monophasic or biphasic, what was the orientation of the coil (e.g. 45 degrees to midline)? Was TMS delivered at 120% of RMT over PMd during TMS-EEG assessment?

Section 2.2 – The targeting of premotor cortex is not clear to me from the description in the methods. The authors imply that the MRI (e.g. neuronavigation) was used to determine PMd stimulation site. However, they then state:

“First, the left motor cortex was identified on T1-weighted individual MRIs using a neuronavigation system (Localite, LTD), and the resting motor threshold (RMT) was assessed following international guide-lines [15]. The RMT expressed as % of Maximum Stimulator Output (%MSO) did not significantly differ between the two groups (HC: 51.0 ± 5.38; ECSCZ: 55.75 ± 10.44; p = 0.1305 Wilcoxon rank-sum test). Then, the left premotor cortex was targeted, and the intensity of stimulation was set to 120% of the RMT, with a stimulation angle orthogonal to the left superior frontal gyrus. The accurate location of the final targeting spot was confirmed using a freely available brain atlas based on Montreal Neurological Institute (MNI) coordinates [16]”

It seems like the authors might have used motor evoked potentials elicited by TMS (120% of RMT) at sites anterior to the motor cortex hotspot to determine the site of stimulation for PMd for the TMS-EEG collection. However, more detail is needed over the exact procedure used. Further, what muscle was used to determine RMT? However, this is not clear. Anatomy (MNI coordinate) was then used to confirm the chosen stimulation site was in indeed in the PMd.

Section 2.2 – The authors delivered 150 single pulses with a jittered interstimulus interval (0.4 to 0.6 Hz or ~every 2s) during continuous EEG recording. Can the authors confirm that the stimuli did not have a differential cumulative effect over the 150 trials across group. Studies over motor cortex typically use a much slower rate of stimulation (5-7 s) to reduce the probability of cumulative plasticity-like effects.

Results

1st paragraph - What are “off-target sensory artifacts”, what is the significance of the effective suppression of these?

Figure 1 B/C - Why were TEPs not analyzed? Looking at the traces provided in Figure 1, there are qualitative differences in the amplitudes of the early TEPS of the two representative subjects.

Figure 1B/C – For the ERSP traces, it appears that the onset in ERSP for the healthy subjects precedes the TMS stimulus by upwards of ~20 ms. Should that be possible if the ERSP response is due to the physical effects of the TMS? Is it possible that healthy participants began anticipating the TMS stimulus and the enhanced power was a perceptual effect?

The authors should report the statistics for non-significant correlations between the natural frequency and SAPS/SANS/dose. The authors make a statement in the discussion that such correlation may not have been significant due to sample size. However, with no report of effect size, the reader has no basis to determine the potential validity of this explanation.

Discussion

The discussion is short and it is perhaps telling that the limitations section is as long, if not longer, than the interpretation of the results.

1.       Is a 4 Hz difference in the natural frequency meaningful? Both frequencies are well within the beta range.

2.       What is the potential of TMS-EEG over premotor cortex as a biomarker? Looking at Figure 2, I think it would be difficult to make the claim that the premotor TMS-EEG natural frequency can differentiate at Healthy Control and an early schizophrenic patient with more than chance accuracy. While there is a difference at the group level, Over 9 ECSCZ patients are within the 25th percentile of the HC. Further, 3-5 HC are at or below the median for the ECSCZ group. Therefore, it seems likely that premotor natural frequency would need to be part of a suite of diagnostic assessments. Given their past work and the work by Noda et al. showing correlations between prefrontal cortex TMS-EEG and clinical scores/cognitive assessments, might prefrontal TMS-EEG offer a better potential biomarker?

3.      Several statements also need further expansion and the TMS-EEG SAI work by Noda et al. should be discussed. For example, the authors make the statement:

Finally, unlike other brain areas involved in schizophrenia, such as the prefrontal cortex [38], the premotor cortex is a particularly accessible and easy-to-engage target for TMS-EEG investigations

However, no context is given to this statement. I am guessing this has to do with using MEPs to determine site of TMS. However, several past studies have measured TEPs to prefrontal stimulation in healthy young adults, typical older adults and schizophrenic patients using MRI anatomy or the 10-20 electrode placement system. Since the authors had to obtain MRI to confirm their site of stimulation was PMd, where is the benefit of targeting PMd? Further, the correlations between prefrontal TMS-EEG and clinical/cognitive measures seen by Noda et al. and the authors own past work suggest this easy-to-engage target might not be the best biomarker.

Reviewer 2 Report

The paper "Natural oscillatory frequency slowing in the premotor cortex of early-course schizophrenia patients: a TMS-EEG study" presented EEG-TMS  results from people early-course schizophrenia and demonstrated lower "natural frequencies" compared with healthy controls.  This paper presented interesting results that could have implications for using these reduced frequencies as biomarkers for schizophrenia.

The paper is well-written, and the introduction provides sufficient background and clear aims. It would be great, however if the authors could expand on the finding of the lowered natural frequencies, not just from themselves but other authors too, to build solid reasoning for the study even if this is not necessarily in schizophrenia brains but other disorders.   

For the methods section, a little more detail on the TMS experiment (section 2.2) or maybe change the presentation of this section to make the procedure more understandable- separated by paragraphs would be useful for people in the field. Why were low frequencies chosen? Were there differences for high frequencies? Why left cortex? (maybe discuss the limitations of this too?)

It was discussed well. One comment which may be of interest is the overlap of healthy control (about n=7) with lower natural frequencies, it'll be interesting if this could be discussed as the purpose is potentially that lower frequencies are a biomarker.

Reviewer 3 Report

The topic is interesting but some modifications are required to be clarified.

-In the abstract please unify the fonts because the font of sentence ‘’… the hypothesis of a deficit’’ to the end is different from the beginning.

- the space between ‘’2. MATERIALS AND METHODS’’ and ‘’2.1’’ is empty. please explain what is going to be explained in the next subsections.

-please explain in section ‘’2.1. participants’’ if subjects signed a letter of consent for collaboration and if ethical committee were held for the working with participants.

-please first introduce the abbreviations, before using them, for example, ECSCZ is not introduced.

- In ‘’ 2.2. Procedure’’, it is recommended to mention, under what standard data is recorded 10-10 or 10-20?

-Subsection ‘’2.3. Data Analysis’’ and ‘’2.2. Data Analysis ‘’ is written in one paragraph, it is confusing for the reader to follow the points. Please divide it into multiple paragraphs. Also, the same problem exist in the other parts.

- please mention that did authors used programing for applying the ICA algorithm and the filters? please explain the details of preprocessing and processing sections. If authors used predefined algorithms or software also, please explain it.

-It would be more professional to have a conclusion in the text. Also, the analysis in the results in the discussion part was not satisfiable. please explain it from other points of view

Round 2

Reviewer 3 Report

The modification is satisfiable and can be published in the current format.